# Interventricular Septum Thickness for the Prediction of Coronary Heart Disease and Myocardial Infarction in Hypertension Population: A Prospective Study

**DOI:** 10.3390/jcm11237152

**Published:** 2022-12-01

**Authors:** Yiqing Yang, Zhao Li, Xiaofan Guo, Ying Zhou, Ye Chang, Hongmei Yang, Shasha Yu, Nanxiang Ouyang, Shuang Chen, Guozhe Sun, Yu Hua, Yingxian Sun

**Affiliations:** Department of Cardiology, The First Hospital of China Medical University, 155 Nanjing North Street, Heping District, Shenyang 110001, China

**Keywords:** interventricular septum thickness, left ventricle remodeling, coronary heart disease, myocardial infarction, hypertension

## Abstract

The aim of the present study was to evaluate the prognostic value of interventricular septum thickness (IVSd) on the incidence of cardiovascular diseases. Based on the general population in Northeast China, 10,349 participants were successfully followed up for echocardiography over a median follow-up time of 4.66 years, among which 4801 were hypertensive. Coronary heart disease (CHD) and myocardial infarction (MI) incidence were followed up. Cox proportional hazards models were used to estimate the association of the baseline IVSd with adverse outcomes. IVS hypertrophy increased incident rates of CHD and MI compared with normal IVSd in the overall population and in the female sex-stratification group. In males, IVS hypertrophy had parallel increase rates of CHD (all *p* < 0.05). Kaplan–Meier analysis showed that IVS hypertrophy could predict CHD and MI incidence and CHD-free and MI-free survival. Multivariable Cox analysis revealed that IVS hypertrophy was correlated with CHD incidence (HR = 1.155, 95% CI = 1.155–2.861, *p* = 0.01) and MI incidence (HR = 2.410, 95% CI = 1.303–4.458, *p* = 0.005). In women, IVS hypertrophy was independently associated with CHD and MI incidence (all *p <* 0.05). Our prospective cohort study illustrated that IVS hypertrophy detected by echocardiography has a prognostic significance for CHD and MI. Therefore, the early detection of IVSd should be conducted to avoid adverse outcomes in further clinical practice.

## 1. Introduction

Essential hypertension (EH) is one of the most common cardiovascular diseases. EH can be defined as an elevation in blood pressure for an unexplained reason that increases the risk of brain, heart, and kidney events [1]. Hypertension is a long-course disease with many complications. It has a high rate of disability and mortality. It not only seriously affects patients’ quality of life, but also causes a huge burden on society. Therefore, we should take effective control of blood pressure levels. For example, early intervention therapy to prevent early-target organ damage caused by hypertension, and to reduce the incidence of coronary heart disease (CHD) and myocardial infarction (MI), as well as other cardiovascular events, should be implemented.

Left ventricle hypertrophy (LVH) is a pathological process caused by the chronic cardiac stress load of EH [2]. However, structural changes in LVH in some hypertensive patients are manifested in the interventricular septum, which is called isolated ventricular septum hypertrophy (IVSH) [3]. IVSH is a common cardiac structural change in the early stage of hypertension because the ventricular septum is more sensitive to catecholamine stimulation than the left posterior wall and right ventricle [4]. The degree of IVSH caused by hypertension is positively correlated with blood pressure level. The higher the blood pressure level, the more obvious the degree of IVSH in patients. Since the interventricular septum thickness (IVSd) is easier to measure and to obtain than left ventricular mass by Doppler ultrasound, IVSd is often used as an early sensitivity index to evaluate LVH in patients with EH.

In this study, we aimed to explore the role of IVS, regarded as the categorical variable, on the prediction of the incidence and prognostic value of CHD and MI in the total and sex-stratified analyses in the Northeast China Rural Cardiovascular Health Study (NCRCHS).

## 2. Materials and Methods

### 2.1. Study Population

The cohort was from the Northeast China Rural Cardiovascular Health Study (NCRCHS), and that study’s design, as well as inclusion and exclusion criteria, were explained in our previous work [5]. In brief, a total of 11,956 participants aged ≥ 35 years old were recruited from Dawa, Zhangwu, and Liaoyang counties in Liaoning province between 2012 and 2013. The cohort was designed using a multi-stage, randomly stratified, and cluster-sampling scheme. In 2015 and 2017, the study population were invited to attend a follow-up study. In total, 10,700 participants consented and were qualified to participate. A total of 10,349 subjects (86.6%) finished at least one follow-up visit. Since we studied the population with hypertension, we found that 4801 of the 10,349 subjects were hypertensive. In the normal IVSd group, there were 4419 participants, and the number of participants with IVS hypertrophy was 382. None of the patients received antihypertensive therapy.

### 2.2. Data Collection

At baseline, detailed information about the participants’ demographic characteristics, lifestyle factors, and disease and medical history was acquired by interviewing the participants using a standardized questionnaire.

Smoking and drinking conditions were defined as current use. Weight and height were measured with participants wearing lightweight clothing and by taking off their shoes. Family history of hypertension and family history of cardiovascular disease were self-reported and were verified by medical records. Body mass index (BMI) was calculated as weight (kilograms) divided by the square of the height (m^2^), and obesity was defined as BMI ≥ 28 kg/m^2^ [6]. Blood pressure was measured using a standardized electronic digital sphygmomanometer (HEM-907; Omron, Tokyo, Japan), and participants were asked to rest for more than 5 min before measurements were taken. The final blood pressure was the mean of all three measurements. The definition of hypertension was considered as systolic blood pressure (SBP) ≥ 140 mm Hg and/or diastolic blood pressure (DBP) ≥ 90 mm Hg and/or the use of antihypertensive drugs [7]. The participants fasted for 12 h, and their blood samples were collected in the morning. Low-density lipoprotein cholesterol (LDL-C), high-density lipoprotein cholesterol (HDL-C), fasting plasma glucose (FPG) and other blood routine examination indexes were analyzed enzymatically. Low HDL-C was defined as HDL-C < 1.03 mmol/L (40 mg/dL). High LDL-C was defined as LDL-C ≥ 4.16 mmol/L (160 mg/dL) [8]. Diabetes was defined as FPG ≥ 7.0 mmol/L and/or the use of drug therapy for diabetes [9].

### 2.3. Echocardiography Method

Echocardiography was performed according to the previous study [10]. In brief, the sonographers used a 3.0 MHz transducer to conduct a Doppler echocardiograph (Vivid; GE Healthcare, Boston, MA, USA) with participants in the dorsal position. Two-dimensional images were detected according to the guidelines [11]. Aortic dimensions (AoD) (12.8~27.0 mm in male, 12.4~25.0 mm in female); left atrial size (LA size), specifically the antero-posterior LA diameter (LAD) (23.5~38.7 mm in male, 22.0~36.8 mm in female); interventricular septal thickness (IVSd) (6.4~11.4 mm in male, 5.6~11.6 mm in female); left ventricular end-diastolic diameter (LVEDD) (38.4~54.0 mm in male, 36.7~49.7 mm in female); left ventricular end-systolic diameter (LVESD) (22.6~38.6 mm in male, 20.8~35.4 mm in female); and left ventricular posterior wall thickness at end-systolic (LVPWd) (6.3~11.1 mm in male, 5.5~10.3 mm in female) were measured in the parasternal long-axis view [12]. The left ventricular ejection fraction (LVEF) (52.6~76.2 mm in male, 52.8~77.2 mm in female) [12] was calculated according to modified Simpson’s rules [13]. Estimated by Teichholz equations, the LV end-diastolic volume (LVEDV) was (mL) = LVEDD^3^ × 7.0/(2.4 + LVEDD) (45.9~127.5 mL in male, 37.7~106.7 mL in female), and the LV end-systolic volume (LVESV) was (mL) = LVESD^3^ × 7.0/(2.4 + LVESD) (12.4~50.0 mL in male, 8.4~43.6 mm in female) [12]. LVEF = [(LVEDV − LVESV)/LVEDV] × 100%. The mitral E-peak velocity (VE peak) (0.44~1.18 m/s in male, 0.48~1.30 m/s in female) was the mitral valve maximum peak velocity of rapid filling during early diastole. The mitral A-peak velocity (VA peak) (0.28~1.06 m/s in male, 0.27~1.17 m/s in female) [12] was the mitral valve maximum peak velocity of atrial contraction during late diastole. The aortic valve velocity (AVV) (0.79~1.65 m/s in male, 0.84~1.74 m/s in female) was measured by an aortic velocity–time integral during a single respiratory cycle. IVS hypertrophy was defined as IVSd > 1.14 cm for males and as > 1.06 cm for females according to the Guidelines for the Measurement of Echocardiography in Chinese Adults [14]. The echocardiography results for all of the participants were analyzed by three experienced ultrasound doctors. If they disagreed with each other on the results, we asked an additional two ultrasound doctors to make the final decision. All procedures complied with the guidelines from the American Society of Echocardiography [15]. 

### 2.4. Definition of Clinical Outcomes

During the follow up, the participants’ clinical information was collected using a face-to-face standardized questionnaire administered by village doctors. Over the median follow-up time of 4.66 years, the effective follow-up numbers were 10,349, and the follow-up rate was 86.6%. In this cohort, we defined the endpoint events as coronary heart disease (CHD) and myocardial infarction (MI), which were defined as new cases diagnosed during the follow-up period. The history of CHD and MI was excluded. CHD was defined as a diagnosis of hospitalized angina, hospitalized myocardial infarction, any revascularization procedure, or CHD-related death [16]. MI was defined as myocardial ischemia and myocardial necrosis in a clinical setting. Meeting any of the following criteria can result in the diagnosis of myocardial infarction: cardiac biomarkers (e.g., troponin) rise and/or fall combined with at least one of the following: ischemic symptoms; new ischemia or the development of pathological Q waves in ECG changes; imaging evidence of activity loss of myocardium or newly abnormal partial wall motion; sudden death; and pathological findings of an acute myocardial infarction [17]. If a participant received any of the above diagnoses or died, their clinical information was collected, including their medical records and death certificate. Finally, the end-point assessment committee would read and judge the variety of disease.

### 2.5. Statistical Analysis

All data were analyzed by R (version 3.6.1; https://www.rstudio.com/, accessed on 1 March 2020.). Continuous variables were reported as values ± standard deviations if they had a normal distribution and were used as a nonparametric test if they had an abnormal distribution. Categorical variables were reported as numbers and percentages. This was dependent on the data type, so we chose the t-test, analysis of variance (ANOVA), nonparametric test, or χ2-test. Kaplan–Meier estimates were utilized to calculate the cumulative incidence of CHD and MI for each group, and the log-rank test was used to compare the differences between each group. Multivariable Cox proportional hazard models were adopted to calculate hazard ratios (HRs) with 95% confidence intervals (CIs) for the associations between IVS hypertrophy and CHD and MI. All of the tests were two-tailed, and *p* < 0.05 was considered statistically significant.

### 2.6. Ethics Statement

The study was approved by the Ethics Committee of China Medical University (Shenyang, China, ID: 2018194) and in keeping with the Declaration of Helsinki.

## 3. Results

### 3.1. Baseline Information of Study Participants

According to the standard of classification, there were 4419 (92.0%) subjects with normal IVSd and 382 (8.0%) subjects with IVS hypertrophy. Apart from a family history of CVD and smoking, there were significant differences in the baseline clinical and echocardiographic characteristics between the normal IVSd group and IVS hypertrophy group (all *p* value < 0.05), as shown in Table 1.

### 3.2. Incidences of CHD and MI in Study Population

Over a median follow-up of 4.66 years, 146 subjects (3.0%) developed CHD, and 67 subjects (1.4%) experienced MI. The incidences of overall CHD and MI were significantly elevated in participants with IVS hypertrophy than in the normal IVSd group (all *p <* 0.001). On the basis of sex, the study population was ulteriorly subdivided. Compared with the normal IVSd group, the incidence of CHD in both the male and female populations with IVS hypertrophy were statistically higher. However, only female subjects with IVS hypertrophy had an elevated incidence of CHD and MI (all *p <* 0.05), as in shown in Table 2.

### 3.3. IVS Hypertrophy for the Prediction of CHD and MI during Follow Up

Kaplan–Meier survival estimates showed that subjects with IVS hypertrophy had higher cumulative CHD and MI incidence than those with normal IVSd in the participants overall (Figure 1). There was an association between IVS hypertrophy and CHD incidence in males and females, and similar tendencies were discovered for MI incidence in females (all *p <* 0.05) (Figure 2). As is shown in Figure 3, IVS hypertrophy was significantly related to worse CHD-free survival in both males and females, and comparable trends were observed for MI-free survival in females (all *p <* 0.05).

Table 3 shows the relationship between IVS hypertrophy and poor outcomes of CHD and MI using Cox proportional hazard models. Before adjustment, IVS hypertrophy was associated with CHD incidence and MI incidence in the total population and in the female group, whereas in males, IVS hypertrophy was correlated with CHD incidence (all *p <* 0.01). After adjustments for sex, age, DM, BMI, Current_smoke, Current_drink, FamilyHis, Mean_SBP, Mean_DBP, GLU, and LDL_C, HDL_C as appropriate, we conducted a new analysis. In the total population, IVS hypertrophy was correlated with CHD incidence (HR = 1.155, 95% CI = 1.155–2.861, *p* = 0.01) and MI incidence (HR = 2.410, 95% CI = 1.303–4.458, *p* = 0.005). Furthermore, we stratified the subgroups by sex and found that in the female subgroup, IVS hypertrophy was independently associated with CHD incidence (HR = 1.870, 95% CI = 1.062–3.292, *p* = 0.030) and MI incidence (HR = 3.063, 95% CI = 1.449–6.474, *p* = 0.003). We carefully examined our population and found that there were four people who had acute or chronic renal insufficiency. Considering that these illnesses can result in hypertension, we eliminated these four people from our cohort and conducted a new Cox proportional hazard model analysis as shown in Appendix A. However, there was a tiny difference between this model and the former model comprising 4801. Our analysis showed that the reason for this was that the constituent ratio of the four people in the total population was small. The four people comprised two men and two women, and none of them had either IVS hypertrophy or cardiovascular events such as MI or CHD.

## 4. Discussion

In this large Chinese population cohort, we showed that, in patients with hypertension, a significantly higher incidence of CHD and MI occurred in the IVS hypertrophy group compared to in the normal IVS group in the overall and female groups. This study demonstrated that abnormal IVS may be an independent prognostic factor for these adverse outcomes. Our study may help with the early intervention of hypertension-associated cardiovascular events in clinical practice.

Some studies have reported that IVS was associated with an increased risk of adverse outcomes. For example, in a hypertensive population study, researchers found that diastolic left ventricular filling dysfunction and cardiac dysrhythmias were increased among patients with isolated IVS hypertrophy and normal LV mass [3]. In a study of 51 apparently healthy pilots, Harpaz et al. demonstrated that isolated IVS hypertrophy was associated with hypertension, even with normal LV mass condition. This study revealed that IVS hypertrophy patients identified by echocardiography should be closely monitored for hypertension [18]. Grossman et al. also revealed that IVS predicts future systolic hypertension in young healthy pilots [19]. In a prospective elderly cohort study of 5888 participants, Gardin et al. found that the incidence of congestive heart failure (CHF) in women had significantly greater IVSd, whereas in men, the incidence of CHD, CHF, stroke, and all-cause mortality had significantly greater IVSd [20]. Apostolakis et al. demonstrated that moderately or severely abnormal IVSd was an independent predictor of stroke and all-cause mortality in atrial fibrillation patients. This may indicate that we should pay close attention to IVS hypertrophy regression in this population [21]. In patients undergoing aortic valve replacement, Straten et al. found that increased IVSd was a predictor of late mortality [22]. 

Given that IVSd was reported to be associated with adverse outcomes in a large number of studies, there were some explanations that may help us to understand our findings. First, IVS thickening may influence hemodynamics and activate the neurohumoral mechanism during the pathophysiology process, showing the possible links between IVS hypertrophy and adverse clinical outcomes. Second, hypertension is a risk factor of cardiovascular diseases [23,24,25]. It plays an important role in the progression of cardiovascular diseases. Third, IVS thickening causes LV hypertrophy, which has been shown to be a significant predictive factor of adverse cardiovascular outcomes [26,27]. In summary, IVSd evaluated by the echocardiography method as a key risk factor has significant implications for improving risk stratification and prediction. There are other causes of IVS hypertrophy: hypertrophic obstructive cardiomyopathy (HOCM), in which IVS is not proportional to the posterior wall of the left ventricle; the ratio of IVS, where the posterior wall of the left ventricle was ≥1.3, and there was a pressure difference in the left ventricular outflow tract [28]; chronic renal failure: these patients always have moderate-to-severe hypertension, and four of ten patients showed asymmetric septal hypertrophy (ASH) typical of hypertrophic cardiomyopathy (HCM) on echo [29]; and diabetes: Aman et al. reported that in spite of efforts to achieve good glycemic control during pregnancy, both fat mass and cardiac septal thickness were increased in newborn infants of women with T1DM and GDM. This condition seems to be related to glycemic control and fetal hyperinsulinemia [30].

We found that IVS is more significant in females rather than males as a predictor of CHD and MI. There are some explanations for this: first, estrogen may protect premenopausal woman from CHD risk to some extent. Endothelium-dependent vasodilation was greater in premenopausal than in postmenopausal women [31]. The beneficial vasodilatory effect of estrogen has also been found in the coronary arteries [32]. However, in postmenopausal women, estrogen levels are decreased, and this protective effect is weakened. Second, Wenger pointed out that although chest pain was the most common symptom of MI for both sexes, more women were inclined to describe polypnea and extreme fatigue, which may lead to their delayed diagnosis and therapy [33]. Third, the long-term use of oral contraceptives was also a risk factor for cardiovascular diseases such as hypertension [34] and abnormal coagulation mechanisms [35]. Fourth, compared to men, middle-aged and elderly women were more likely to have decreased physical fitness, obesity, and overweight [36], which are also risk factors for CHD and MI. Last, cardiac rehabilitation was underused for women [37]. Therefore, educational messages from public health departments are needed to target racial and ethnic minority women regarding their presentation of acute MI and the vulnerability of women.

Our study had some clinical significance. Now that IVSd has provided the prediction value of CHD and MI in adults with hypertension, we should pay attention to IVSd and avoid remodeling in the early stage. Angiotensin-converting enzyme inhibitors (ACEIs), a kind of drug used in the treatment of hypertension, have been reported to successfully reverse LV hypertrophy and myocardial fibrosis in spontaneously hypertensive rats [38,39]. ACEIs play pharmacological roles by reducing LV mass, regressing LV hypertrophy, decreasing vascular atherosclerosis, and improving vascular compliance [40]. Therefore, ACEI should be used in the early stage of IVS thickening to avoid LV remodeling and a series of cardiovascular events in further clinical practice.

However, there were some limitations in our study. First, this cohort was from a single-center study and was mainly located in the Northeast of China, and some patients were excluded in the follow-up due to the absence of ultrasonic data, which may have caused selection bias. Second, our study focused on the prognostic value of IVSd in hypertensive patients, so caution should be exercised when applying the results to the general population. Third, the follow-up time may have been too short to determine the presence of cardiovascular events. Therefore, a multi-center study and a longer follow-up time are necessary to further validate our conclusions.

## 5. Conclusions

Our study found that hypertension patients with IVS hypertrophy were more likely to experience an elevated incidence of CHD and MI than those with normal IVSd. IVSd was associated with a parallel increased risk of incident CHD and MI in the female group. Overall, intervening measures should be put into practice to reduce further adverse cardiovascular events in the hypertension population.

## Figures and Tables

**Figure 1 jcm-11-07152-f001:**
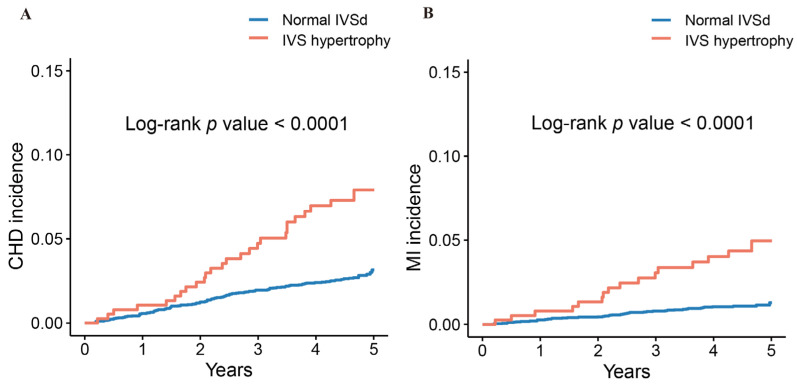
Kaplan–Meier survival curves for CHD incidence (**A**) and MI incidence (**B**) in overall population according to IVSd. CHD, coronary heart disease; MI, myocardial infarction; IVSd, interventricular septum thickness; IVS Hypertrophy, Interventricular septum hypertrophy.

**Figure 2 jcm-11-07152-f002:**
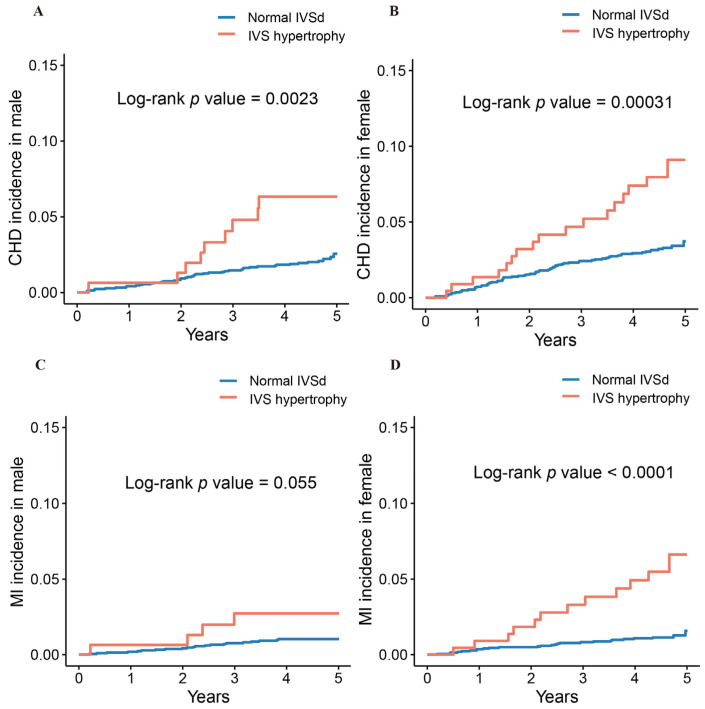
Kaplan–Meier survival curves for CHD incidence in males (**A**) and females (**B**); MI incidence in males (**C**) and females (**D**). CHD, coronary heart disease; MI, myocardial infarction; IVSd, interventricular septum thickness; IVS Hypertrophy, Interventricular septum hypertrophy.

**Figure 3 jcm-11-07152-f003:**
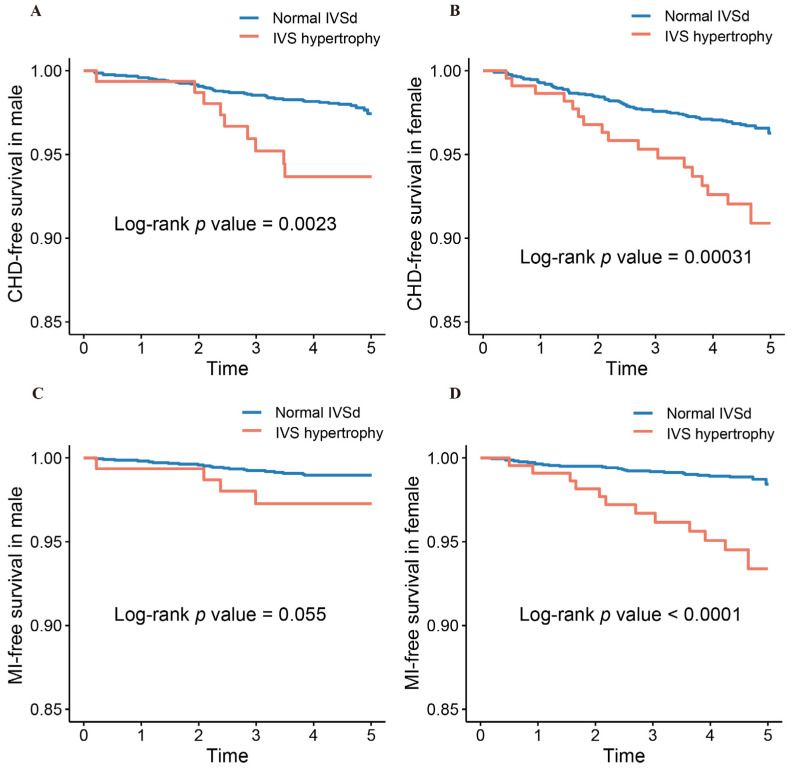
Kaplan–Meier survival curves for CHD-free survival in males (**A**) and females (**B**); MI-free survival in males (**C**) and females (**D**). CHD, coronary heart disease; MI, myocardial infarction; IVSd, interventricular septum thickness; IVS hypertrophy, interventricular septum hypertrophy.

**Table 1 jcm-11-07152-t001:** Baseline information of study participants.

Variables	Normal IVSd	IVS Hypertrophy	*p* Value
	(n = 4419)	(n = 382)	
Female, n (%)	2243 (50.8)	225 (58.9)	0.003
Male, n (%)	2176 (49.2)	157 (41.1)	0.003
Age (years)	56.91 ± 10.04	59.94 ± 10.30	<0.001
Diabetes (%)	256 (5.8)	39 (10.2)	0.001
Family_HT (%)	1146 (25.9)	122 (31.9)	0.013
Family_CVD (%)	1296 (29.3)	122 (31.9)	0.311
Smoking (%)	1575 (35.6)	120 (31.4)	0.109
Drinking (%)	1120 (25.3)	66 (17.3)	0.001
BMI (kg/m^2^)	25.44 ± 3.53	27.12 ± 4.01	<0.001
SBP (mmHg)	157.91 ± 18.47	173.18 ± 25.46	<0.001
DBP (mmHg)	88.33 ± 10.58	94.26 ± 14.61	<0.001
AoD (cm)	2.28 ± 0.32	2.35 ± 0.38	<0.001
LA size (cm)	3.42 ± 0.39	3.74 ± 0.48	<0.001
IVSd (cm)	0.89 ± 0.10	1.26 ± 0.57	<0.001
LVEDD (cm)	4.76 ± 0.42	4.94 ± 0.54	<0.001
LVESD (cm)	3.16 ± 0.44	3.29 ± 0.52	<0.001
LVPWd (cm)	0.88 ± 0.23	1.07 ± 0.12	<0.001
LVEF (%)	62.48 ± 3.60	61.53 ± 4.03	<0.001
VEpeak (cm/s)	71.02 ± 24.55	64.75 ± 19.87	<0.001
VApeak (cm/s)	81.07 ± 21.02	87.31 ± 21.20	<0.001
AVV (cm/s)	120.00 ± 20.35	125.16 ± 28.85	<0.001
FPG (mmol/L)	6.12 ± 1.79	6.45 ± 2.40	<0.001
HDL_C (mmol/L)	1.44 ± 0.41	1.33 ± 0.34	<0.001
LDL_C (mmol/L)	3.11 ± 0.87	3.20 ± 0.91	<0.001

IVSd = interventricular septum thickness; IVS Hypertrophy = Interventricular septum hypertrophy; Family_HT = family history of hypertension; CVD = cardiovascular disease; Family_CVD = family history of CVD; BMI = body mass index, SBP = systolic blood pressure; DBP = diastolic blood pressure; AoD = aortic dimension; LA size = left atrial size; IVSd = interventricular septal thickness; LVEDD = left ventricular end-diastolic diameter; LVESD = left ventricular end-systolic diameter; LVPWd = left ventricular posterior wall thickness at diastole; LVEF = left ventricular ejection fraction; VEpeak = mitral E-peak velocity; VApeak = mitral A-peak velocity; AVV = aortic valve velocity; FPG = fasting plasma glucose; LDL-C = low-density lipoprotein cholesterol; HDL-C = high-density lipoprotein cholesterol.

**Table 2 jcm-11-07152-t002:** Incidences of CHD and MI in study population.

Outcomes	Overall (n = 4801)		Normal IVSd (n = 4419)		IVS Hypertrophy (n = 382)		*p* Value
	N (%)	Rate per 1000	N (%)	Rate per 1000	N (%)	Rate per 1000	
		Person-Years (95%CI)		Person-Years (95%CI)		Person-Years (95%CI)	
Overall							
CHD	146 (3.0)	7.046 (5.949–8.286)	120 (2.7)	6.262 (5.192–7.487)	26 (6.8)	16.686 (10.9–24.448)	<0.001
MI	67 (1.4)	3.208 (2.486–4.074)	51 (1.2)	2.641 (1.967–3.473)	16 (4.2)	10.147 (5.8–16.477)	<0.001
Male							
CHD	55 (1.1)	5.424 (4.086–7.06)	46 (1.0)	4.849 (3.55–6.468)	9 (2.4)	13.777 (6.3–26.154)	<0.001
MI	26 (0.5)	2.551 (1.667–3.738)	22 (0.5)	2.308 (1.446–3.494)	4 (1.0)	6.068 (1.653–15.537)	0.1486
Female							
CHD	91 (1.9)	8.599 (6.924–10.558)	74 (1.7)	7.647 (6.004–9.6)	17 (4.5)	18.785 (10.943–30.077)	<0.001
MI	41 (0.9)	3.834 (2.751–5.201)	29 (0.7)	2.966 (1.987–4.26)	12 (3.1)	13.076 (6.757–22.841)	<0.001

IVSd = interventricular septum thickness; IVS Hypertrophy = Interventricular septum hypertrophy; CHD = coronary heart disease; MI = myocardial infarction.

**Table 3 jcm-11-07152-t003:** IVS hypertrophy for the prediction of CHD, MI and stroke during follow up.

Subgroups	CHD			MI		
	Cases/Noncases	Incidence HR (95% CI)	*p* Value	Case/Noncases	Incidence HR (95% CI)	*p* Value
*Before Adjustment*						
Overall						
Normal IVSd	120/4299	Reference	/	51/4368	Reference	/
IVS hypertrophy	26/356	2.757 (1.802–4.219)	<0.001	16/366	4.157 (2.357–7.332)	<0.001
Male						
Normal IVSd	46/2130	Reference	/	22/2154	Reference	/
IVS hypertrophy	9/148	2.893 (1.414–5.917)	0.004	4/153	2.732 (0.938–7.96)	0.065
Female						
Normal IVSd	74/2169	Reference	/	29/2214	Reference	/
IVS hypertrophy	17/208	2.558 (1.507–4.344)	<0.001	12/213	4.951(2.497–9.817)	<0.001
*After Adjustment*						
Overall						
Normal IVSd	120/4299	Reference	/	51/4368	Reference	/
IVS hypertrophy	26/356	1.818 (1.155–2.816)	0.01	16/366	2.410 (1.303–4.458)	0.005
Male						
Normal IVSd	46/2130	Reference	/	22/2154	Reference	/
IVS hypertrophy	9/148	1.873 (0.875–4.008)	0.106	4/153	1.639 (0.516–5.208)	0.402
Female						
Normal IVSd	74/2169	Reference	/	29/2214	Reference	/
IVS hypertrophy	17/208	1.870 (1.062–3.292)	0.030	12/213	3.063 (1.449–6.474)	0.003

Adjusted for Sex, Age, DM, BMI, Current_smoke, Current_drink, FamilyHis, Mean_SBP, Mean_DBP, GLU, LDL_C, HDL_C as appropriate. Abbreviations: CHD, coronary heart disease; MI, myocardial infarction; IVSd, interventricular septum thickness; IVS hypertrophy, interventricular septum hypertrophy.

## Data Availability

Data can be obtained from the corresponding author upon reasonable request.

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
