# Peer review of "Interventricular Septum Thickness for the Prediction of Coronary Heart Disease and Myocardial Infarction in Hypertension Population: A Prospective Study"

_jcm, 2022, doi:10.3390/jcm11237152_

Round 1

Reviewer 1 Report

General Comments

This is a large cohort of patient with hypertension and IVS hypertrophy. The results showed significant role of IVS hypertrophy as risk factor of CHD and MI. The grammar should be improved to better show the significance of the study results.

Specific Comments

1. No explanation on why the IVS is more significant in female as a predictor of CHD and MI. Please provide it.

2. Please recheck the grammar thoroughly. Please have a native English speaker to finalize it.

3. The figures 1 and 2 of Kaplan Meier. Please choose 1 either event free survival or incidence graphs.

Reviewer 2 Report

In this article, Yang et all. present a study that investigated the risk of coronary heart disease and myocardial infarction in hypertension population. The study design and results were presented. The study is interesting. However, some points need to be addressed.

1.     Regarding your study population, please explain if patients had any renal disorders. Especially when looking at hypertension, renal dysfunction or renal stenosis should be investigated and kept in mind.

2.     Please give further information about other conditions your patients had during your study e.g., pulmonal hypertonia, copd etc.

3.     Please give further details about medication your patients take that might increase cardiovascular risk.

4.     Please add laboratory parameters.

5.     Tab. 1: Please also include female patients in your table.

6.     Tab.1: Please quantify smoking status by using information like e.g., pack years

7.     Please explain in more detail how drinking condition was defined. How many grams of alcohol per week/ month was considered as current use?

8.     Your discussion needs to address why women have a higher risk than men for CHD

 and MI.

9.     Please discuss other causes of IVS hypertrophy.

10.  Please discuss in more detail why MI is the first manifestation in patients with IVS. Did the patients not have any symptoms or treatment before?

11.  Please just show the incidence time or incidence free time as both is too much information.

12.  Please discuss why small differences in echocardiography data is significant in your study while there is no clinical difference e.g., LVEF just differs 1% between the groups.

Reviewer 3 Report

It is well known that  IVS plays an important role in cardiac function and the detailed evaluation of  IVS can predict CAD and myocardial infarction.

I have some questions:

1.     Did you analyzed paradoxical septal motion  among all patients?

2.     How many patients had basal septal hypertrophy?

3.     Did you observed any ventricular tachycardia before MI

4.     Did you use longitudinal strain of IVS as a echocardiography marker ?

5.     Did you analyzed all patients with IVS hypertrophy by doing noninvasive diagnostic of CAD?
